# The quality and safety of *Rhodiola rosea* supplements on the U.S. market: An analysis of biomarkers, heavy metals, and pesticide residues

Steffen Porwollik, Mahtab Jafari 🔵 *

School of Pharmacy & Pharmaceutical Sciences, University of California, Irvine, California, United States of America

* mjafari@uci.edu

## Abstract

*Rhodiola rosea* supplements have become popular in the U.S., with a $3.4 billion market and an annual growth of about 10% in sales in the past few years. While the health benefits of this plant have been evaluated in many scientific studies, the potential differences in quality of these botanical products on the U.S. market have not been studied in detail. Using reversed-phase ultra-performance liquid chromatography, we determined the concentrations of the biomarker molecules, rosavins and salidroside, in a small but representative sample of *R. rosea* dietary supplement products commercially available in the U.S. Concentrations of rosavins and salidroside ranged from 0.01% to 3.08% and 0.07% to 2.91%, respectively, including substantial aberrations from advertised biomarker amounts. One product showed an undisclosed likely addition of synthetic salidroside. We also assessed heavy metal contaminations via inductively coupled plasma mass spectrometry and pesticide contents by gas chromatography-mass spectrometry or liquid chromatography-mass spectrometry in these botanical products. While pesticide levels were below detection limits, all seven tested capsular products had trace amounts of arsenic, cobalt, and lead. Two of these contained notably elevated levels of cobalt and arsenic, where follow-up arsenic speciation would be required to verify whether they remain within minimal risk levels established by the U.S. Department of Health and Human Services. Overall, these results underscore the need for more stringent quality control in herbal supplements containing *R. rosea* available in the U.S.

## Introduction

*Rhodiola rosea* (a plant in the Crassulaceae family) is also commonly known as golden root, rose root, or Arctic root, and is a widely used adaptogenic plant recognized for its potential health benefits. The plant has a long history of use in traditional

**Data availability statement:** All relevant data are within the manuscript and its Supporting Information files.

**Funding:** The author(s) received no specific funding for this work.

**Competing interests:** The authors have declared that no competing interests exist.

medicine, particularly in Scandinavian countries, Russia, China, and other regions with cold climates. It has been utilized for centuries to enhance physical endurance, reduce fatigue, and promote resilience against environmental stressors [1,2].

*Rhodiola rosea* has been reported to improve exercise performance [3], to help with stress management [4,5], and to improve mental health [6]. It has also displayed anti-aging properties, increasing lifespan and improving healthspan, in various animal model systems, such as fruit flies, silkworms, and rats [7–10]. Due to these purported effects, *R. rosea* has been incorporated into a variety of increasingly popular dietary supplements. Global sales of *R. rosea*-containing supplements have experienced significant growth, and in the United States, these supplements have experienced a compound annual growth rate exceeding 10% between 2019 and 2024 [11].

*Rhodiola rosea* naturally thrives in high-altitude, dry, and cold regions, including the Arctic, the mountains of Central Asia, and parts of North America and Europe. It grows in rocky, nutrient-poor soils and exhibits remarkable adaptability to harsh environmental conditions. Due to increasing commercial demand, concerns about overharvesting and habitat degradation of wild *R. rosea* are voiced by environmental scientists [12,13]. Although *R. rosea* is currently not included in the *International Union for Conservation of Nature*'s *Red List of Threatened Species*, the plant has been added to the list of protected species by the *Convention on International Trade in Endangered Species* in late 2022 [13], making sure that its international trade is regulated to facilitate sustainability in its harvesting from the wild.

*Rhodiola rosea* contains a diverse array of phytochemicals, with approximately 140 different compounds identified, including glucosides, phenols, flavonoids, and monoterpenes [5,14]. Among the glucosides, the phenylpropanoid glycosides rosarin, rosin, and rosavin, collectively called rosavins, as well as salidroside, are recommended as key biomarker molecules and reported in HPLC-based analyses verifying the quality of this plant's extracts [15,16]. However, the concentration of these compounds varies within different plant parts and between male and female specimens [17]. Additionally, environmental factors such as altitude, soil composition, and climate conditions significantly influence the phytochemical composition of *R. rosea*, further complicating standardization efforts [18]. Nevertheless, standardized pharmacopeial references for *R. rosea* exist in many countries, including Russia, Australia, and Northern European nations. The U.S. Pharmacopeia (USP Certificate for catalog number 1602580, *Rhodiola rosea* root and rhizome dry extract) includes a chromatographic profile specifying rosavins (rosin, rosarin, and rosavin) and salidroside as the main biomarkers of the plant, and the USP Herbal Medicines Compendium indicates lower limits of salidroside (0.08%), and rosavins (0.3%) in dried *R. rosea* roots and rhizomes [19]. Some of the adaptogenic effects of the plant, including its antioxidant properties, have been attributed to these biomarker molecules [20,21], but the beneficial effects cannot be consistently recapitulated with isolated rosavins or salidroside in laboratory settings. Other molecules, such as quercetin-glycosides, have also been identified as contributors [22]. Therefore, the adaptogenic properties are likely caused by the synergistic activity of at least some of the >100 organic compounds of the plant.

While dietary supplements are not subjected to the same pre-market scrutiny and regulatory oversight as pharmaceuticals, their manufacturing regulations are governed by 21 CFR Part 111, a regulation set by the U.S. Food and Drug Administration (FDA), which covers specifications for identity, purity, strength, composition, and limits on contaminants. However, surveillance of these products, post-market, often simply relies on reports of adverse events and subsequent label modifications [23], and the quality of the *R. rosea* supplements in the U.S. has not been analyzed in detail. Previous studies have identified inconsistencies in *R. rosea* extract composition across the European market [24–27], where widespread adulteration with other *Rhodiola* species or synthetic compounds has been reported, raising safety concerns and doubts about the products' efficacy. Such fluctuations in quality, active ingredient concentrations, and labeling accuracy abound in the dietary supplement market, and have also been shown for CBD-containing melatonin gummies [28] and immune health dietary supplements [29] sold in the U.S.

This study takes a snapshot of the quality of *R. rosea* supplements currently available on the U.S. market. Specifically, the study assesses the content of the biomarker molecules, rosavins and salidroside, of the plant as well as potential contaminations with heavy metals and pesticides of ten *R. rosea* products commercially available in the U.S. Heavy metal exposure can have dramatic detrimental effects on human health, particularly on kidney function [30] while pesticides may cause chronic health conditions, including asthma, cancer, or diabetes [31]. By analyzing these *R. rosea* products on the U.S. market, this study appraises their safety and variability, informing regulatory agencies, healthcare professionals, and, most importantly, consumers.

## Materials & methods

### Sample selection and preparation for analysis

The selection of ten *Rhodiola rosea* supplements from the U.S. market was based on brand diversity, the products' cost (where we aimed to cover a broad range), and availability. In addition, products needed to be manufactured at separate sites. All capsules and tinctures were purchased from Amazon in 2024.

Capsular products were homogenized by creating a composite of capsule fill material, which was then sampled. Fill material from 3–10 capsules per product was used for the analysis. Prior to injection into chromatography systems, capsule and tincture samples were centrifuged and filtered using a 0.45-micrometer filter.

### Chemical analysis

All chemical analyses were completed at the Eurofins Scientific testing facilities in Brea, CA, and Madison, WI, following all applicable procedure-specific industry standards, as required for accredited testing laboratories in the U.S. For the quantification of rosavins and salidroside, a standardized chromatography-based method was employed [19]. Samples were extracted in a mixture of phosphoric acid and methanol and analyzed via reversed-phase ultra-performance liquid chromatography (UPLC) equipped with a photodiode array detector on a Waters™ Acquity I-Class PLUS UPLC System, employing external standards for calibration.

For heavy metal ion analysis, methods 2015.01, 2011.19, and 993.14, as published by AOAC INTERNATIONAL, were followed, with minor modifications [32]. Briefly, samples were digested with nitric acid ($HNO_3$) using a microwave digestion system. Detection was conducted via inductively coupled plasma mass spectrometry (ICP-MS) on Agilent 7900 ICP-MS Systems, where digested samples were compared to calibration standards of known concentrations.

For pesticide determination, compound levels were extracted and processed essentially as described in the AOAC INTERNATIONAL method 2007.01 [32], following the QuEChERS (Quick, Easy, Cheap, Effective, Rugged, and Safe) strategy [33,34]. The protocol involves acetonitrile extraction, partitioning, and dispersive solid-phase extraction clean-up, followed by compound detection via gas chromatography-mass spectrometry (GC/MS) or liquid chromatography mass spectrometry (LC/MS) on Agilent 6400 Triple Quadrupole LC/MS, Agilent 7890 GC/MS, and Agilent 8890 GC/MS Systems.

## Quality assurance

Certified reference materials and calibration standards were used in all chromatography assays to ensure data accuracy. Instruments were routinely subjected to quality control determinations to validate results. Method validation parameters, including limits of detection, quantification, and recovery rates, were incorporated where available to ensure analytical robustness.

For rosavins and salidroside quantification, the limits of quantitation (LOQs) were 50 µg/g (ppm), with a precision of approximately 5% relative standard deviation. For pesticide determination, LOQs were typically at or below 0.01 µg/g for most analytes, except for azinphos-methyl, endosulfans, and heptachlor endo-epoxide (0.02 µg/g) and pyrethrum (0.2 µg/g). For heavy metal ion analysis, LOQs varied depending on the matrix but were typically 5 ppb for cadmium, mercury, and lead, and 10 ppb for arsenic and cobalt.

## Results

### Biomarker molecules

The concentrations of key biomarker molecules in the ten commercial *Rhodiola rosea* products are illustrated in Fig 1 and Table 1 and reveal substantial variability.

For rosavins, measured concentrations ranged from 0.01% to 3.08%, with rosavin contributing between half and two-thirds, rosarin between one-fifth and one-third, and rosin accounting for the remainder. The three tinctures exhibited similar rosavins content, ranging from 0.13% to 0.28%. Only one of these, tincture 1, declared target concentrations of these biomarkers on their label, which closely matched our measurements. Among the seven capsule products, rosavins concentrations varied greatly. Three capsules advertised a target concentration of 3% rosavins, where two closely

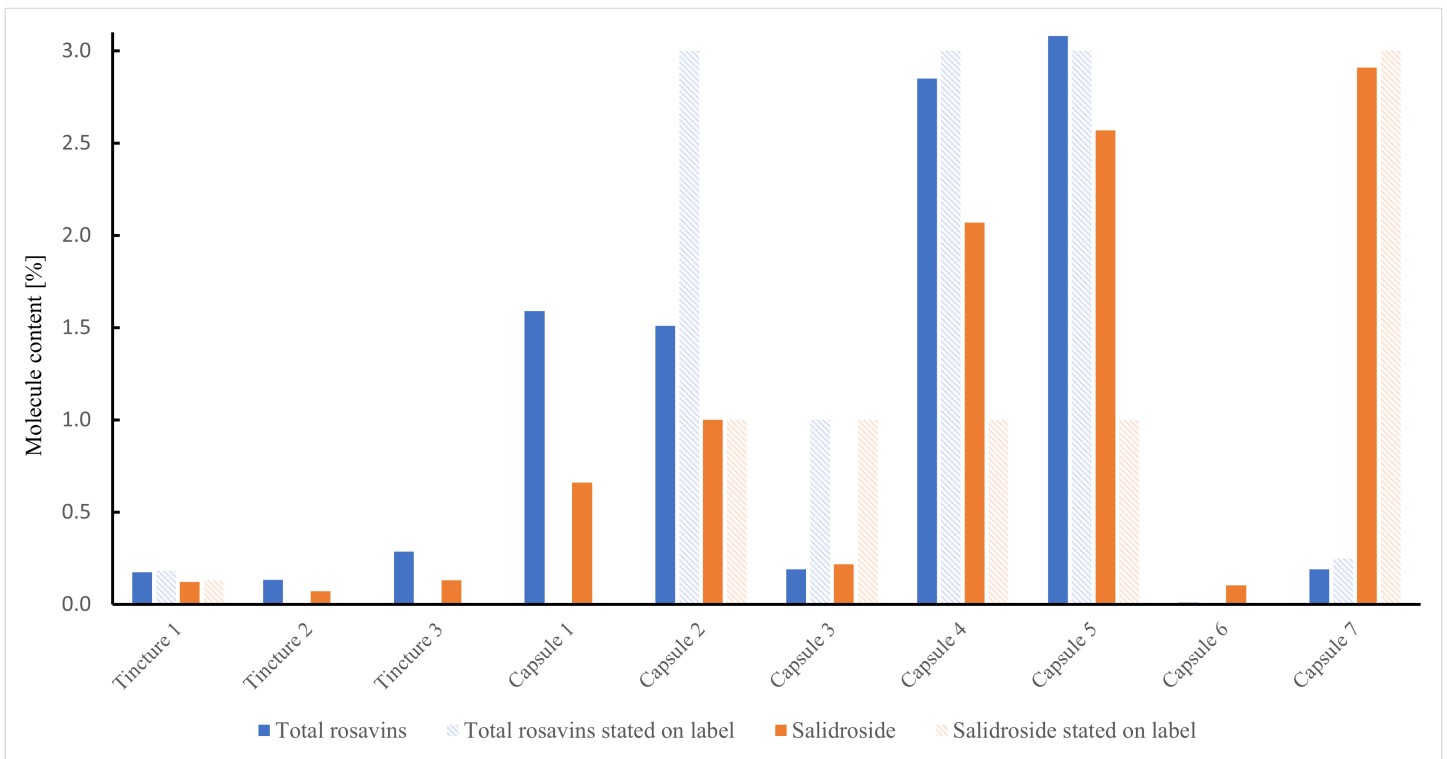

**Fig 1. Contents of rosavins and salidroside in 10 commercially available *Rhodiola rosea* products in the U.S. Where known, advertised concentrations of these biomarker molecules are also shown.**

**Table 1. Biomarker, heavy metal, and pesticide contents of ten *Rhodiola rosea* products available on the U.S. market. * = Biomarker content is significantly lower than advertised. N.D. = not detected, ppb = parts per billion.**

| Sample Name | Rosavin [%] | Rosarin [%] | Rosin [%] | Total Rosavins (advertised on product label) [%] | Salidroside (advertised on product label) [%] | Arsenic [ppb] | Cadmium [ppb] | Cobalt [ppb] | Lead [ppb] | Mercury [ppb] | Pesticides |
|---|---|---|---|---|---|---|---|---|---|---|---|
| Capsule 1 | 1.010 | 0.370 | 0.211 | **1.590** | **0.661** | 22.8 | 34.7 | 46.4 | 75.1 | < 5 | N.D. |
| Capsule 2 | 0.754 | 0.507 | 0.246 | **1.510** *(3.0)** | **1.000** *(1.0)* | 54.4 | < 5 | 50.8 | 11.6 | < 5 | N.D. |
| Capsule 3 | 0.118 | 0.049 | 0.023 | **0.191** *(1.0)** | **0.217** *(1.0)** | 21 | < 5 | 18 | 9 | < 5 | N.D. |
| Capsule 4 | 1.680 | 0.726 | 0.437 | **2.850** *(3.0)* | **2.070** *(1.0)* | 159 | 22 | 164 | 88 | < 5 | N.D. |
| Capsule 5 | 1.900 | 0.705 | 0.474 | **3.080** *(3.0)* | **2.570** *(1.0)* | 393 | 11 | 733 | 85 | < 5 | N.D. |
| Capsule 6 | 0.006 | 0.002 | 0.002 | **0.011** | **0.104** | 65.8 | < 5 | 40.8 | 43.8 | < 5 | N.D. |
| Capsule 7 | 0.100 | 0.061 | 0.030 | **0.191** *(0.25)* | **2.910** *(3.0)* | 59.7 | < 5 | 106 | 55.2 | 10.9 | N.D. |
| Tincture 1 | 0.083 | 0.057 | 0.033 | **0.173** *(0.183)* | **0.122** *(0.131)* | < 10 | < 5 | < 10 | < 5 | < 5 | N.D. |
| Tincture 2 | 0.087 | 0.030 | 0.017 | **0.134** | **0.071** | < 10 | < 5 | < 10 | < 5 | < 5 | N.D. |
| Tincture 3 | 0.194 | 0.064 | 0.029 | **0.286** | **0.130** | < 10 | < 5 | < 10 | < 5 | < 5 | N.D. |

matched that target while capsule 2 fell about two-fold short. Furthermore, capsule 3, labeled as containing 1% rosavins, had measured concentrations nearly five times lower than declared. One product, capsule 6, contained only 0.01% rosavins, suggesting an unexpectedly low *R. rosea* extract content.

For salidroside, a similarly wide concentration range was observed. The three tinctures contained salidroside levels between 0.07% and 0.13%, whereas capsule concentrations varied from 0.10% to 2.9%. Four capsule products advertised a 1% salidroside content. Three of these (capsules 2, 4, and 5) either matched or exceeded that target, while capsule 3 missed it by almost five-fold. Capsule 7, labeled as containing 3% salidroside, had a measured concentration of 2.9%, closely matching its claimed value.

## Heavy metals and pesticides

We quantified ions from five heavy metals—arsenic, cadmium, cobalt, lead, and mercury—in all ten products. Because most heavy metal ions do not readily extricate from plant material into liquid extracts [35], the tinctures were not expected to contain problematic amounts of cadmium, cobalt, lead, or mercury ions. As expected, none of the three tinctures had detectable levels of these metal ions, nor of arsenic ions. However, the seven capsule products exhibited considerable variation in contamination levels (Table 1, Fig 2). Mercury ions were only detected in capsule 7 (10.9 ppb), while cadmium ions were present in three capsules (11.0–34.7 ppb). Ions from the remaining three metals were detected in all capsules: arsenic (21–393 ppb), cobalt (18–733 ppb), and lead (9–88 ppb). While capsule 4 had elevated amounts of arsenic and cobalt ions, capsule 5 exhibited the greatest levels of ions of these two heavy metals, with concentrations over 6.5-fold and over 14-fold higher, respectively, than the median values detected across all capsules.

Furthermore, we explored whether the products contained any of the 70 pesticides listed in the U.S. Pharmacopeia (USP) chapter <561>. All quantified pesticide compounds—including DDT, chlorpyrifos, heptachlor, and hexachlorobenzene—remained below detection limits. However, USP <561> focuses on contaminants historically relevant to herbal products and botanical pharmaceuticals, and does not include glyphosate or neonicotinoids, two pesticide classes of growing concern in agricultural and environmental contexts.

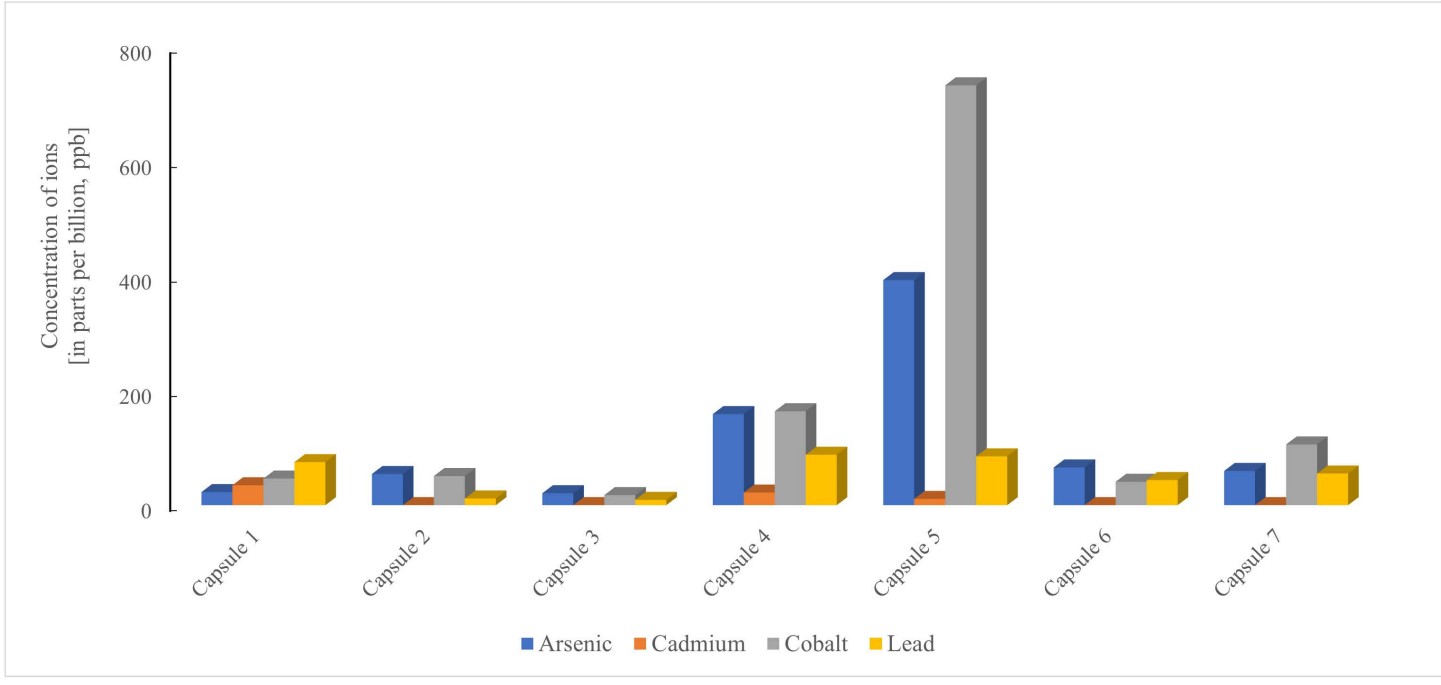

**Fig 2. Heavy metal content in seven commercial capsule preparations of *Rhodiola rosea*.**

## Discussion

The ten *Rhodiola rosea* supplements analyzed in this study exhibited substantial variability in biomarker molecule content. While this is the first study exploring *R. rosea* products available in the U.S., our observations align with prior studies of dry extracts commercially available in Europe [26], where the lowest salidroside concentrations were approximately nine times higher than the lowest concentrations detected in our study among capsular *R. rosea* products. While rosavins levels in some European samples exceeded 5%, we observed low amounts of this biomarker in several samples. A low rosavins content has been linked to adulteration with other *Rhodiola* species that lack this biomarker compound, such as *Rhodiola crenulata* [25]. Our findings identified three capsular products in which rosavins concentrations were lower than their corresponding salidroside amounts, raising concerns of potential adulteration. In one case, capsule 7, the low rosavins content coincided with a 15-fold higher salidroside concentration, suggesting adulteration, blending with other *Rhodiola* species, or manipulation of biosynthetic pathways in *Rhodiola* cell culture suspensions [36]. Although the product label accurately reported the salidroside concentration, it did not disclose the addition of synthetic salidroside or the presence of other *Rhodiola* species. Notably, one tested capsular product advertised a rosavins concentration of 1%, yet measured levels were approximately five times lower. In contrast, two other products claiming a 3% rosavins content contained levels very close to the target concentration, while one other product exhibited rosavins at 1.5%, a two-fold lower concentration than the stated content.

Each milliliter of the tincture products tested in this study is reported to be made from 300–500 mg of dried root of *R. rosea*. These tinctures did not contain pesticides and heavy metals, but displayed lower concentrations of rosavins and salidroside than most other products surveyed here and elsewhere [19,25]. However, the biological activity and adaptogenic properties of *R. rosea* may not be solely dependent on its biomarker molecule content. Over the past two decades, we have tested the biological activity of *R. rosea* products in fruit flies and mice [8,9,37,38]. Our recent lifespan assays in *Drosophila melanogaster* demonstrated comparable life-extending effects between *R. rosea* extracts with lower

concentrations of rosavins and an *R. rosea* extract with a higher concentration of rosavins (S1 File). Whether this equivalence extends to other purported benefits of the plant, such as enhanced energy, mental health support, stress reduction, and exercise performance, remains to be determined. Although salidroside and rosavins are frequently cited as bioactive compounds, their physiological bioavailability, synergy with other plant compounds, or the presence of other molecules in commercially developed *R. rosea* extracts may affect their functional profile *in vivo*. In fact, total *R. rosea* extracts are known to perform better than isolated biomarkers [18].

Heavy metal contamination is a longstanding concern in herbal supplements. A global survey found that approximately 30% of 1,700 tested samples of 86 herbal medicines contained at least one heavy metal above regulatory limits [39], and over a quarter of 2,300 herbal products sold in 37 countries showed signs of adulteration [40], a practice that occurs in even higher proportions in popular supplements such as ginkgo and black cohosh [41], and in 29% of herbal products sold in the US [42]. In our study, all seven tested capsular products contained trace amounts of arsenic, cobalt, and lead ions. While most levels were low, capsule 5 contained approximately 200 ng of arsenic and 360 ng of cobalt per serving (500 mg), while capsule 4 had about 40 ng of arsenic and cobalt per serving (250 mg). The U.S. Department of Health and Human Services established a minimal risk level (MRL) for oral cobalt ingestion at 30 ng/kg/day [43]. Notable arsenic compounds include inorganic arsenic, monomethylarsonic acid (MMA), and dimethylarsinic acid (DMA). For inorganic arsenic, MRLs are 5 ng/kg/day for acute-duration ingestion (14 days or less) and 0.3 ng/kg/day for chronic-duration ingestion (> 1 year) [44]. For MMA, MRLs are higher (100 ng/kg/day and 10 ng/kg/day for acute-duration and chronic-duration ingestion, respectively), and the established MRL for chronic ingestion of DMA is 20 ng/kg/day [44]. Our detection procedure does not include arsenic speciation, and we are therefore unable to verify whether capsules 4 and 5 exceed established MRLs or not.

None of the ten tested products contained pesticide residues above our detection limits. However, given the small sample size, we cannot exclude the presence of such contaminants in other *R. rosea* products available on the market. Regular, comprehensive safety assessments would be instrumental in addressing public concerns regarding the quality and safety of these supplements. Enhanced regulatory oversight and standardized quality control measures are essential to ensure consumer safety and product efficacy. Enforcing Good Manufacturing Practices, ensuring transparent supply chains, and implementing third-party verification could further safeguard consumers while supporting the conservation of this valuable medicinal plant in its natural habitat.

In summary, this study represents the first investigation of biomarker molecule variability and contamination levels in commercially available *R. rosea* products in the U.S. While biomarker compound concentrations varied considerably, the impact of this variability on product efficacy remains unclear. More critically, we detected (occasionally considerable) heavy metal contamination in all capsular products. Consumers should therefore be vigilant about product sourcing, manufacturer transparency, ingredient traceability, and independent third-party validations. The results of our survey urge more frequent and widespread testing and improved quality control strategies for U.S. *R. rosea* products, to ensure both product quality and consumer safety.

## Supporting information

**S1 File. Lifespan extension of *Drosophila melanogaster* with *Rhodiola rosea* extracts containing different biomarker concentrations.**
(XLSX)

## Acknowledgments

We are grateful to Eurofins Scientific, USA, for their technical and scientific support and critical review of the manuscript. This study was funded by an unrestricted gift from the Kay Family Foundation at Irvine, CA.

## Author contributions

**Conceptualization:** Mahtab Jafari.

**Data curation:** Steffen Porwollik, Mahtab Jafari.

**Formal analysis:** Steffen Porwollik, Mahtab Jafari.

**Funding acquisition:** Mahtab Jafari.

**Investigation:** Mahtab Jafari.

**Methodology:** Steffen Porwollik, Mahtab Jafari.

**Project administration:** Mahtab Jafari.

**Resources:** Mahtab Jafari.

**Supervision:** Mahtab Jafari.

**Validation:** Mahtab Jafari.

**Writing – original draft:** Steffen Porwollik, Mahtab Jafari.

**Writing – review & editing:** Steffen Porwollik, Mahtab Jafari.

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
