## [Decision Letter · Decision Letter 0]

30 Jul 2025

Dear Dr. Jafari,

Thank you for submitting your manuscript to PLOS ONE. After careful consideration, we feel that it has merit but does not fully meet PLOS ONE’s publication criteria as it currently stands. Therefore, we invite you to submit a revised version of the manuscript that addresses the points raised during the review process.

The manuscript requires some revisions before it can be resubmitted for peer review.

We look forward to receiving your revised manuscript.

Kind regards,

Mahmoud W. Yaish, Ph.D.

Academic Editor

PLOS ONE

Reviewers' comments:

Reviewer's Responses to Questions

**Comments to the Author**

1. Is the manuscript technically sound, and do the data support the conclusions?

Reviewer #1: Yes

Reviewer #2: Yes

2. Has the statistical analysis been performed appropriately and rigorously?

Reviewer #1: N/A

Reviewer #2: N/A

3. Have the authors made all data underlying the findings in their manuscript fully available?

Reviewer #1: Yes

Reviewer #2: Yes

4. Is the manuscript presented in an intelligible fashion and written in standard English?

Reviewer #1: Yes

Reviewer #2: Yes

Reviewer #1: Adulteration of botanical ingredients and their products is a big concern for the botanical dietary supplement and related industries. Therefore information about the quality and authenticity of commercial products is important to gather. For this manuscript, the authors have used standard analytical test methods to evaluate the levels of phenolic marker compounds, some heavy metals, and pesticides in 10 commercial samples labeled to contain Rhodiola rosea root/rhizome.

The text is easy to read and follow, and the experimental part is well done. There are a few minor suggested edits, and suggestions for additional publications that may be helpful to include. But overall, the manuscript can be accepted after a few modifications.

Line 40: Please add the botanical family

Line 45: “is considered an emerging botanical”… I am not familiar with this term. Presumably the authors mean to write that rhodiola is a botanical with growing consumer interest, but this would be an odd way to state this.

Line 50: During which time period did the AGR exceed 10%?

Lines 56 and 57: According to one source (Bejar, 2017), “in some geographical areas, the two most frequently used species, R. crenulata and R. rosea, are becoming vulnerable or at-risk (one source uses the terms “threatened” and “critically endangered” when referring to specific areas).” Hence, I don’t think the sentence that R. rosea is not classified as endanegers is accurate. It depends on the location.

Line 67: What do you mean by USP Catalog No: 1602580? Please indicate which USP monograph you refer to.

Line 75: “surveillance of these products, post-market, is sporadic at best,…How do you know that? Please cite some references to support that post-market surveillance is sporadic. I am not aware of such data, but there is a lot of prejudice about what is going on in the industry.

Line 89: How did you determine “brand-diversity”. Please specify what criteria you used.

Line 90: Well-regarded distributors…like whom? Please specify which distributors you refer to. And what are your criteria for well-regarded?

Line 197: Two additional papers on the frequency of adulteration of botanicals:

- Ichim MC. The DNA-based authentication of commercial herbal products reveals their globally widespread adulteration. Front Pharmacol. 2019;10:1227. 10.3389/fphar.2019.01227

- Orhan N, Gafner S, Blumenthal M. Estimating the extent of adulteration of the popular herbs black cohosh, echinacea, elder berry, ginkgo, and turmeric – its challenges and limitations. Nat Prod Rep. 2024;41(10):1604-1621. https://doi.org/10.1039/d4np00014e

Other publications worth considering:

- Bejar E, Upton R, Cardellina II JH. Adulteration of rhodiola (Rhodiola rosea) rhizome and root extract. Botanical Adulterants Prevention Bulletin. Austin TX, ABC-AHP-NCNPR Botanical Adulterants Prevention Program. 2017;1-8. http://doi.org/10.59520/bapp.bapb/PenB1664

- Gemma S, Multari G, Turco L, Gallo FR. A densitometric HPTLC method for the simultaneous quali-quantitative determination of rosavin and salidroside in Rhodiola rosea L. based commercial products: A quick screening by HPTLC-based fingerprint and antioxidant evaluation. J Liq Chromatogr Rel Techn. 2025. 10.1080/10826076.2025.2462263

Figures 1 and 2 need to be submitted as high-resolution graphs.

Reviewer #2: Introduction:

1. add full botanical name of Rodiola rosea

2. any information on the mechanism of action is missing

3. it will be beneficial to add information about the negative effect of the potential contaminations.

Materials and methods:

1. It will be beneficial to add if the product sample was a convinience sample or you used any other criteria. When were the supplements purchased?

2. Why did you choose 7 capsule products and 3 tinctures?

Discussion:

1. it will be very beneficial to add a comment on the biological activity of the supplements based on the different biomarker molecules content.

**Do you want your identity to be public for this peer review?** For information about this choice, including consent withdrawal, please see our Privacy Policy

Reviewer #1: **Yes: ** Stefan Gafner

Reviewer #2: No

---

## [Author Response · Author response to Decision Letter 1]

22 Aug 2025

August 22, 2025

Ref: Submission PONE-D-25-18554; The quality of Rhodiola rosea supplements on the U.S. market: An analysis of biomarker molecule, heavy metal, and pesticide contents

Dear Dr. Yaish,

We sincerely thank both reviewers for their careful assessment of our manuscript and the many useful suggestions that they made. Their efforts have improved our manuscript substantially. We have carefully considered all comments and have made appropriate revisions to the manuscript. For each reviewer comment, we provide a detailed response and indicate any corresponding changes made to the manuscript.

As requested, we have modified the manuscript format to adhere to the PLoS One guidelines. Both authors agree to full publication of all underlying data related to this manuscript. Regarding our previous use of “data not shown”, we have now prepared a supplementary file that contains the underlying data in support of the relevant statement.

After further review and considering the reviewers’ comments, we have retained anonymity of all products and removed any identifying reference (e.g., specific sources such as Alaskan Rhodiola rosea) to keep the focus on the analytical findings.

We have carefully considered all comments and have made appropriate revisions to the manuscript. Please see our point-by-point response below. Reviewers’ comments are copied and listed in black, and our responses are listed in blue. For each reviewer comment, we provide a detailed response and indicate any corresponding changes made to the manuscript.

We hope these revisions address the reviewers’ comments and demonstrate our commitment to strengthening the scientific contribution of this study.

Sincerely,

Mahtab Jafari, PharmD

Professor of Pharmaceutical Sciences

Director, The UC Irvine Center for Healthspan Sciences

University of California, Irvine - School of Pharmacy and Pharmaceutical Sciences

Reviewer #1

Adulteration of botanical ingredients and their products is a big concern for the botanical dietary supplement and related industries. Therefore information about the quality and authenticity of commercial products is important to gather. For this manuscript, the authors have used standard analytical test methods to evaluate the levels of phenolic marker compounds, some heavy metals, and pesticides in 10 commercial samples labeled to contain Rhodiola rosea root/rhizome.

The text is easy to read and follow, and the experimental part is well done. There are a few minor suggested edits, and suggestions for additional publications that may be helpful to include. But overall, the manuscript can be accepted after a few modifications.

REPLY: Thank you so much for this positive feedback.

Line 40: Please add the botanical family

REPLY: The botanical family Crassulaceae was added, as suggested.

Line 45: “is considered an emerging botanical”… I am not familiar with this term. Presumably the authors mean to write that rhodiola is a botanical with growing consumer interest, but this would be an odd way to state this.

REPLY: Thank you for pointing this out. We have removed this phrase.

Line 50: During which time period did the AGR exceed 10%?

REPLY: We have now clarified the CAGR by specifying that “…in the United States, these supplements have experienced a compound annual growth rate exceeding 10% between 2019 and 2024 [11].”

Lines 56 and 57: According to one source (Bejar, 2017), “in some geographical areas, the two most frequently used species, R. crenulata and R. rosea, are becoming vulnerable or at-risk (one source uses the terms “threatened” and “critically endangered” when referring to specific areas).” Hence, I don’t think the sentence that R. rosea is not classified as endanegers is accurate. It depends on the location.

REPLY: Thank you so much for this thoughtful comment. We agree with your comment and have addressed this ambiguity by stating that “…R. rosea is currently not included in the International Union for Conservation of Nature’s Red List of Threatened Species.”

Line 67: What do you mean by USP Catalog No: 1602580? Please indicate which USP monograph you refer to.

REPLY: We apologize for this ambiguity. We referred to the US Pharmacopeial Convention’s Certificate for catalog number 1602580, Rhodiola rosea root and rhizome dry extract. We have now added the full name to the text.

Line 75: “surveillance of these products, post-market, is sporadic at best,…How do you know that? Please cite some references to support that post-market surveillance is sporadic. I am not aware of such data, but there is a lot of prejudice about what is going on in the industry.

REPLY: Thank you for your question and suggestion. We were indeed imprecise. We have now modified the statement and say that “…surveillance of these products, post-market, often simply relies on reports of adverse events and subsequent label modifications [23].” The supporting reference is Li W, Wertheimer A. 2023. Innov Pharm. PMCID: PMC10686678.

Line 89: How did you determine “brand-diversity”. Please specify what criteria you used.

REPLY: We assembled a panel of products, where each product was from a different brand and produced at a different manufacturing site. We now add this information to our selection criteria.

Line 90: Well-regarded distributors…like whom? Please specify which distributors you refer to. And what are your criteria for well-regarded?

REPLY: Thank you for pointing this out. This characterization was not based on published criteria and has therefore been removed.

Line 197: Two additional papers on the frequency of adulteration of botanicals:

- Ichim MC. The DNA-based authentication of commercial herbal products reveals their globally widespread adulteration. Front Pharmacol. 2019;10:1227. 10.3389/fphar.2019.01227

- Orhan N, Gafner S, Blumenthal M. Estimating the extent of adulteration of the popular herbs black cohosh, echinacea, elder berry, ginkgo, and turmeric – its challenges and limitations. Nat Prod Rep. 2024;41(10):1604-1621. https://doi.org/10.1039/d4np00014e

REPLY: Thank you so much, we have now added these references to the manuscript.

Other publications worth considering:

- Bejar E, Upton R, Cardellina II JH. Adulteration of rhodiola (Rhodiola rosea) rhizome and root extract. Botanical Adulterants Prevention Bulletin. Austin TX, ABC-AHP-NCNPR Botanical Adulterants Prevention Program. 2017;1-8. http://doi.org/10.59520/bapp.bapb/PenB1664

- Gemma S, Multari G, Turco L, Gallo FR. A densitometric HPTLC method for the simultaneous quali-quantitative determination of rosavin and salidroside in Rhodiola rosea L. based commercial products: A quick screening by HPTLC-based fingerprint and antioxidant evaluation. J Liq Chromatogr Rel Techn. 2025. 10.1080/10826076.2025.2462263

REPLY: We sincerely thank the reviewer for bringing these references to our attention. We now include the highly relevant Bejar publication in our revised manuscript, as suggested.

Figures 1 and 2 need to be submitted as high-resolution graphs.

REPLY: Thank you for pointing this out. We investigated the resolution of our figures.

The uploaded versions of Figures 1 and 2 have high-resolution. It appears that their resolution changes once embedded in the PDF file. We communicated this issue with the PLOS One editorial staff, and they suggested the following:

“Since reviewers and the Academic Editor will have access to your File Inventory, they can download the files in their original resolution as necessary while evaluating the manuscript. If the manuscript is accepted, the figures or images in question will be linked in their full resolution. “

Reviewer #2

Introduction:

1. add full botanical name of Rodiola rosea

REPLY: Thank you. We have added the botanical family of Rhodiola rosea (Crassulaceae) to the manuscript text.

2. any information on the mechanism of action is missing

REPLY: Thank you for pointing this out. We had refrained from such a statement because the exact mechanism of action of R. rosea is still under investigation. Most previous studies centered on possible effects of the biomarker molecules salidroside and rosavins, which are thought to contribute to the adaptogenic effects of the plant. However, not all of R. rosea’s biological activities can be attributed to these biomarker molecules, and the benefits are likely imparted by the synergistic contribution of several of the over 140 organic compounds that have been identified in the plant. We very much look forward to future studies characterizing the impact of the various compounds on the plant’s adaptogenic properties. For now, we have added the following text to the manuscript describing the current state of knowledge on the mechanism of action of the plant. “Some of the adaptogenic effects of the plant, including its antioxidant properties, have been attributed to these biomarker molecules [20, 21], but the beneficial effects cannot be consistently recapitulated with isolated rosavins or salidroside in laboratory settings. Other molecules, such as quercetin-glycosides, have also been identified as contributors [22]. Therefore, the adaptogenic properties are likely caused by the synergistic activity of at least some of the > 100 organic compounds of the plant.”

3. it will be beneficial to add information about the negative effect of the potential contaminations.

REPLY: Thank you so much for this comment. We have now expressed the dangers of heavy metal and pesticide contamination with the following section: “Heavy metal exposure can have dramatic detrimental effects on human health, particularly on kidney function [30], while pesticides may cause chronic health conditions, including asthma, cancer, or diabetes [31].” In addition, we also point out that contamination with other Rhodiola species may raise doubts regarding the product’s efficacy.

Materials and methods:

1. It will be beneficial to add if the product sample was a convinience sample or you used any other criteria. When were the supplements purchased?

REPLY: Thank you for pointing out our oversight in characterizing the selection in more detail. All capsules and tinctures were purchased in 2024, and we have now added this information to the text. The samples were bestselling R. rosea products on Amazon.com, and included products in different price ranges. We added this description to the text.

2. Why did you choose 7 capsule products and 3 tinctures?

REPLY: The inclusion of tinctures provided better coverage of the different product types on the U.S. market. We selected more capsules than tinctures because more R. rosea capsule products are sold on the market, compared to tinctures. We estimated that this small sample of products, which included the current online bestsellers in different price ranges, would provide a good snapshot of the quality of the R. rosea products on the U.S. market.

Discussion:

1. it will be very beneficial to add a comment on the biological activity of the supplements based on the different biomarker molecules content.

REPLY: This is a great idea, but it is difficult to provide such an estimate. As outlined above, and mentioned in our discussion, the biological activity of the Rhodiola rosea supplements cannot be reliably based on biomarker molecule content alone, and is, instead, a result of complex (and hitherto incompletely characterized) interactions of the many organic compounds of the plant.

Therefore, we largely refrain from speculating on the biomarkers’ biological activity. However, we included in our discussion that “Our recent lifespan assays in Drosophila melanogaster demonstrated comparable life-extending effects between R. rosea extracts with lower concentrations of rosavins and an R. rosea extract with a higher concentration of rosavins (S1 File). Whether this equivalence extends to other purported benefits of the plant, such as enhanced energy, mental health support, stress reduction, and exercise performance, remains to be determined.”

Rhodiola rosea products are standardized according to their rosavins and salidroside content, which is why we refer to them as biomarker molecules. Our goal was to verify the composition of the products on the market in the U.S., and to estimate their safety by checking for contamination with some of the most important health hazards. A survey of the biological activity of the products or the biomarker molecules was beyond the scope of this study. We wholeheartedly agree that future studies focusing on the biological activity of these products and their biomarkers would be highly informative. These future studies may include assays measuring the biological activities of a standardized amount of each product in cell cultures, as well as physical performance and stress evaluations in laboratory animals.

---

## [Decision Letter · Decision Letter 1]

20 Nov 2025

Dear Dr. Jafari,

Thank you for submitting your manuscript to PLOS ONE. After careful consideration, we feel that it has merit but does not fully meet PLOS ONE’s publication criteria as it currently stands. Therefore, we invite you to submit a revised version of the manuscript that addresses the points raised during the review process.

The manuscript is important and suitable for publication, but it still needs significant improvements in clarity, methodological detail, formatting, and data presentation before it can be accepted.

We look forward to receiving your revised manuscript.

Kind regards,

Mahmoud W. Yaish, Ph.D.

Academic Editor

PLOS ONE

Journal Requirements:

Additional Editor Comments:

The manuscript is important and suitable for publication, but it still needs significant improvements in clarity, methodological detail, formatting, and data presentation before it can be accepted.

Reviewers' comments:

Reviewer's Responses to Questions

**Comments to the Author**

Reviewer #3: (No Response)

2. Is the manuscript technically sound, and do the data support the conclusions?

Reviewer #3: (No Response)

3. Has the statistical analysis been performed appropriately and rigorously?

Reviewer #3: (No Response)

4. Have the authors made all data underlying the findings in their manuscript fully available?

Reviewer #3: (No Response)

5. Is the manuscript presented in an intelligible fashion and written in standard English?

Reviewer #3: (No Response)

Reviewer #3: The manuscript with the title: The quality of Rhodiola rosea supplements on the U.S. market: An analysis of biomarker molecule, heavy metal, and pesticide contents”, the study is valuable, timely, and relevant to public health, pharmacognosy, and dietary supplement quality control. The work is interesting and the work is suitable for the scope of PLOS one. Here are some comments for improving the manuscript.

1- The title could be changed to Quality and safety of Rhodiola rosea supplements on the U.S. market: variability in biomarkers, heavy metals and pesticide residues

2- In the abstract: Add the analytical methods used in the study (e.g., UPLC, ICP-MS, QuEChERS) to improve clarity in the abstract.

3- in the abstract, the detected concentration range is missing, please adda in the abstract.

4- Keywords is missing, authors should add the keywords in the revised version.

5- All abbreviation should be mentioned in the revised manuscript before the introduction section.

6- Line 58, add suitable reference.

7- in materials and methods section: Please specify how many capsules were pooled per product for analysis?

8- The models of instruments such as ICP-MS, should be mentioned in M& M.

9- Most of the references of the methods are missing in the materials and methods.

10- There is no mentioned about the number of replication in the experiments (duplicate or triplicate).

11- The samples which do not meet the USP minimium limits should be highlighted in the table.

12- Figures 1 & 2: If replicates were performed, add error bars.

13- heavy metals should be in the form heavy metals ions.

14- Figures need high resolution.

15- References, some references the journals are abbreviated , other journals in full name, authors should follow the journal instruction.

**Do you want your identity to be public for this peer review?** For information about this choice, including consent withdrawal, please see our Privacy Policy

Reviewer #3: **Yes: ** Sedky Hassan

---

## [Author Response · Author response to Decision Letter 2]

30 Dec 2025

Reviewer #3

The manuscript with the title: The quality of Rhodiola rosea supplements on the U.S. market: An analysis of biomarker molecule, heavy metal, and pesticide contents”, the study is valuable, timely, and relevant to public health, pharmacognosy, and dietary supplement quality control. The work is interesting and the work is suitable for the scope of PLOS one. Here are some comments for improving the manuscript.

Thank you for the kind evaluation of our work.

1- The title could be changed to Quality and safety of Rhodiola rosea supplements on the U.S. market: variability in biomarkers, heavy metals and pesticide residues

Thank you for this suggestion. We have modified the title and incorporated most of the requested changes. Since there was no variability in pesticide residues, we changed the title to “The quality and safety of Rhodiola rosea supplements on the U.S. market: An analysis of biomarkers, heavy metals, and pesticide residues.”

2- In the abstract: Add the analytical methods used in the study (e.g., UPLC, ICP-MS, QuEChERS) to improve clarity in the abstract.

Thank you for this excellent idea. We have now modified the abstract to include these technical details.

3- in the abstract, the detected concentration range is missing, please adda in the abstract.

Thank you for this suggestion. We have added the concentration ranges to the abstract, as suggested.

4- Keywords is missing, authors should add the keywords in the revised version.

While keywords are not requested or shown in PLoS One’s author guidelines, we now provide five of them.

5- All abbreviation should be mentioned in the revised manuscript before the introduction section.

PLoS One’s instructions for authors at https://journals.plos.org/plosone/s/submission-guidelines.com recommend to “[d]efine abbreviations upon first appearance in the text.” We have chosen to follow this guideline in our submission. We have also removed some abbreviations that only occurred once in the manuscript.

6- Line 58, add suitable reference.

We have expanded the text in the manuscript to clarify the way Rhodiola rosea trade regulations were instituted by the Convention on International Trade in Endangered Species, and have added this reference, as suggested by the reviewer.

7- in materials and methods section: Please specify how many capsules were pooled per product for analysis?

Thank you for this suggestion. We have added the following sentence to the Sample selection and preparation section in Materials & methods: “Fill material from 3 – 10 capsules per product was used for the analysis.”

8- The models of instruments such as ICP-MS, should be mentioned in M& M.

We have added the instruments used for the ICP-MS (Agilent 7900), the UPLC (Waters Acquity I-class), the GC/MS (Agilent 7880 and 8890), and the LC/MS protocols (Agilent 6400), as requested.

9- Most of the references of the methods are missing in the materials and methods.

Thank you for pointing out this omission. We have now included additional specific references from the AOAC compendium used in industrial settings for heavy metal and pesticide detection protocols. Please note that these procedures are routine and standardized industrial protocols used in U.S.-accredited and certified laboratories such as Eurofins.

10- There is no mentioned about the number of replication in the experiments (duplicate or triplicate).

Measurements were only performed once, a strategy also applied in similar high-profile publications evaluating biological compounds in dietary products, such as Cohen et al., 2023 (Quantity of melatonin and CBD in melatonin gummies sold in the US. JAMA. 2023;329(16):1401–1402. doi:10.1001/jama.2023.2296) or Crawford et al., 2022 (Analysis of select dietary supplement products marketed to support or boost the immune system. JAMA Netw Open. 2022;5(8):e2226040. doi:10.1001/jamanetworkopen.2022.26040).

The analyses reported in our manuscript were performed by Eurofins Scientific. The technical error in all applied methods was precisely controlled by reference standards sourced from qualified vendors, and measurements were automatically repeated when system suitability criteria were not met. Furthermore, all SOPs and associated documentation were tightly followed in each method, minimizing potential measurement errors.

11- The samples which do not meet the USP minimium limits should be highlighted in the table.

Thank you for this comment. Please note that the USP minimum limits are set for dried plant roots and rhizomes, as indicated in the Introduction. Those limits do not apply to dietary products containing Rhodiola rosea. Therefore, we refrained from implementing the suggested highlights. However, we have now added the advertised biomarker content on the products’ labels (where known) to Table 1, and marked with an asterisk the two capsular products where measured biomarker amounts were significantly lower than advertised.

12- Figures 1 & 2: If replicates were performed, add error bars.

As explained in our response to comment #10, replicate analyses were not performed due to the controlled precision of the measurements. Therefore, no error bars are shown.

13- heavy metals should be in the form heavy metals ions.

Thank you for this comment. We have rephrased the corresponding section in our manuscript (Heavy metals and pesticides in the Results chapter) and altered the Y-axis title in Figure 2 to reflect that the ions of heavy metals were detected in our analyses.

14- Figures need high resolution.

We agree and hope that our original images fulfill these criteria. We tested our graphs on Newgen Art Analysis, as suggested in PLoS One’s Instructions for Authors, and they passed. We are uncertain why the images appear in such low resolution after upload onto the PLoS One server, and we will work with the editor if this problem persists.

15- References, some references the journals are abbreviated , other journals in full name, authors should follow the journal instruction.

Thank you for pointing this out. We have now corrected this oversight.

---

## [Editor Report · Decision Letter 2]

2 Jan 2026

The quality and safety of Rhodiola rosea supplements on the U.S. market: An analysis of biomarkers, heavy metals and pesticide residues

PONE-D-25-18554R2

Dear Dr. Jafari,

We’re pleased to inform you that your manuscript has been judged scientifically suitable for publication and will be formally accepted for publication once it meets all outstanding technical requirements.

Kind regards,

Mahmoud W. Yaish, Ph.D.

Academic Editor

PLOS One

Additional Editor Comments (optional):

The authors have provided satisfactory responses to all reviewer comments and have adequately addressed the identified concerns. The manuscript is now substantially improved, and its scientific merits support acceptance for publication.
---

## [Editor Report · Acceptance letter]

PONE-D-25-18554R2

PLOS One

Dear Dr. Jafari,

I'm pleased to inform you that your manuscript has been deemed suitable for publication in PLOS One. Congratulations! Your manuscript is now being handed over to our production team.

Kind regards,

on behalf of

Dr. Mahmoud W. Yaish

Academic Editor

PLOS One